# Cystinuria in Dogs and Cats: What Do We Know after Almost 200 Years?

**DOI:** 10.3390/ani11082437

**Published:** 2021-08-19

**Authors:** Simona Kovaříková, Petr Maršálek, Kateřina Vrbová

**Affiliations:** 1Department of Animal Protection and Welfare and Veterinary Public Health, Faculty of Veterinary Hygiene and Ecology, University of Veterinary Sciences, 612 42 Brno, Czech Republic; marsalekp@vfu.cz; 2Faculty of Veterinary Medicine, University of Veterinary Sciences, 612 42 Brno, Czech Republic; Katerina.vrbova96@seznam.cz

**Keywords:** cystine urolithiasis, inborn error, mutation, COLA

## Abstract

**Simple Summary:**

Cystinuria, as an inborn error of metabolism, is a problem with worldwide distribution and has been reported in various canine and feline breeds. Transepithelial transport of cystine is mediated by COLA transporter and the mutation in genes coding this transporter may cause cystinuria. Urolithiasis associated with typical clinical signs may be the clinical consequence of cystinuria. The mutation causing cystinuria and the mode of inheritance have been determined only in several canine breeds. This makes cystinuria difficult to control and gradually decreases its prevalence. In cats, cystinuria occurs only rarely.

**Abstract:**

The purpose of this review is to summarize current knowledge on canine and feline cystinuria from available scientific reports. Cystinuria is an inherited metabolic defect characterized by abnormal intestinal and renal amino acid transport in which cystine and the dibasic amino acids ornithine, lysine, and arginine are involved (COLA). At a normal urine pH, ornithine, lysine, and arginine are soluble, but cysteine forms a dimer, cystine, which is relatively insoluble, resulting in crystal precipitation. Mutations in genes coding COLA transporter and the mode of inheritance were identified only in some canine breeds. Cystinuric dogs may form uroliths (mostly in lower urinary tract) which are associated with typical clinical symptoms. The prevalence of cystine urolithiasis is much higher in European countries (up to 14% according to the recent reports) when compared to North America (United States and Canada) where it is approximately 1–3%. Cystinuria may be diagnosed by the detection of cystine urolithiasis, cystine crystalluria, assessment of amino aciduria, or using genetic tests. The management of cystinuria is aimed at urolith removal or dissolution which may be reached by dietary changes or medical treatment. In dogs with androgen-dependent cystinuria, castration will help. In cats, cystinuria occurs less frequently in comparison with dogs.

## 1. Introduction

Cystinuria is an inherited disorder characterized by the impaired reabsorption of cystine in the proximal tubule of the nephron and the gastrointestinal epithelium. The defective transport also involves the other dibasic amino acids ornithine, lysine, and arginine. The term COLA (cystine, ornithine, lysine, arginine) is used for all these amino acids. Nevertheless, only cystinuria results in urolithiasis because these dibasic amino acids are relatively soluble in urine, despite the fact they may reach high concentrations in the urine of affected animals [1]. 

In human, cystinuria was first described by Wollaston in 1810 when he extracted a large cystolith from one of his patients. He named it cystic oxide, because he believed that it had such chemical properties and that the stone had originated from the bladder wall [2]. Although it was later shown not be an oxide nor secreted from the urinary bladder, when isolated, the amino acid was named cystine in recognition of this historical discovery [1]. In 1908, Sir Archibald Garrod suggested cystinuria as a condition that may be an inborn error of metabolism [3] and Dent and Rose hypothesized that cystinuria is an inborn error of cystine transport [4].

In dogs, the presence of cystine uroliths was reported in 1823 and thus cystinuria was the first reported canine inborn error of metabolism [5]. Now, cystinuria is a disease with worldwide distribution and it is known to affect more than 170 canine breeds according to the reports of veterinary urolith analysis laboratories [6]. Cystinuria is not a problem of dogs and cats only. Indeed, it has also been reported in other canids and felids or ferrets [7,8,9,10,11,12]. The purpose of this review is to summarize the current knowledge on cystinuria in dogs and cats.

## 2. Etiopathogenesis of Cystinuria

Cystine is a non-essential sulfur-containing amino acid composed of two molecules of the amino acid cysteine. Cystine is absorbed through the wall of the small intestine and is normally present in low concentrations in plasma. Plasma amino acids are freely filtered at the glomerulus. Under normal conditions, more than 99% of these amino acids are reabsorbed in the proximal renal tubules. The reabsorption of cystine, ornithine, lysine, and arginine is mediated by COLA transporter [13].

### 2.1. COLA Transporter

The COLA transporter (b0,+) was originally thought to be a heterodimer, but is likely a heterotetramer formed from two heterodimers consisting of the rBAT (extracellular heavy chain, encoded by SLC3A1, solute carrier family 3 member 1) and b0,+AT (the light chain, encoded by SLC7A9, solute carrier family 7 member 9) subunits joined by a disulfide bridge [14,15,16]. This system is the main effector of cystine reabsorption in the kidney. The apical transport system b0,+ mediates influx dibasic amino acids and cystine in exchange for neutral amino acids. The subunit b0,+AT has 12 transmembrane domains typical cell membrane transporters and heterodimers with rBAT exclusively to form the COLA amino acid transporter [17]. Mutations in either rBAT or b0,+AT can cause cystinuria. In people, 133 mutations in SLC3A1 and 95 mutations in SLC7A9 have been identified. Reported mutations include nonsense, missense, splicing, frameshifts, and large sequence rearrangements [13].

The same defect is present in the epithelial cells of the small intestine and altered transport of COLA amino acids from gastrointestinal tract occurs [18,19,20], but this is of little consequence as amino acids are primarily absorbed as small peptides. With the exception of lysine, these amino acids are classified as non-essential, and all four of these dibasic amino acids may be absorbed in their dipeptide forms from the gastrointestinal tract [16]. Thus, cystinuria is not associated with protein malnutrition or COLA amino acid deficiency [21]. Nevertheless, it can be associated with disorders of other amino acids. Four English Bulldogs and one Miniature Dachshund with cystinuria were diagnosed with carnitine and taurine deficiency while participating in a clinical trial evaluating dietary management of cystine urolithiasis [22]. In three of these dogs, excessive quantities of carnitine were lost in their urine, but urine excretion of taurine was within the reference range in spite of plasma taurine deficiency. It can be explained by the fact, that cystine is a precursor amino acid and increased renal excretion of cystine may adversely affect taurine synthesis. 

### 2.2. Genetic Aspects

In humans, cystinuria has been classified phenotypically into two types: Type A is caused by defects in SLC3A1 and is inherited in a true autosomal recessive manner, with heterozygotes having a normal urinary excretion of cystine. Type B is caused by SLC7A9 variants and is autosomal dominant with incomplete penetrance, with heterozygotes having a variable degree of cystine hyperexcretion, some within the physiologic range [23]. 

In dogs, the first genetic description of cystinuria was conducted with Newfoundlands and autosomal recessive inheritance was suggested [24]. Later, the C-to-T nonsense mutation in exon 2 of the SLC3A1 gene was described. This mutation acts as an early stop codon, precluding the production of rBAT protein and leading to the loss of b0,+. In that report, cystinuric dogs of other breeds were examined (Swedish Lapphund, Dachshund, German short-haired pointer, Irish Setter, Jack Russel Terrier, Corgi) and either heterozygosity at the SLC3A1 locus or a lack of mutations coding region of the SLC3A1 gene were observed. This finding indicates that cystinuria is genetically heterogenous in dogs [25]. 

Brons et al. (2013) performed a study of mutations associated with cystinuria in various breeds of dogs and a new classification system for canine cystinuria was established according to their results [26]. In cystinuric Labrador Retriever and Labrador mix-breed dogs, the deleted guanine in codon GGC causes a shift of the open reading frame, leading to premature stop codon 41 codons later. This leads to truncation of the rBAT protein to 157 amino acids instead of 784. The early termination probably causes accelerated RNA decay and decreased or no protein synthesis. The mutation was identified in an autosomal recessive form of the disease which is phenotypically and genetically similar to that previously described in Newfoundlands [24,25]. Only homozygous Labrador Retrievers were cystinuric (both males and females, regardless of neuter status) and developed cystine calculi early in life, albeit more frequently and earlier in males. Labrador Retrievers that were heterozygous for this nonsense mutation showed neither signs of the urinary tract disease nor positive nitroprusside test [26]. 

In Australian Cattle dogs with cystinuria, a heterozygous deletion of six bases was found in exon 6 of the SLC3A1 gene. The same homozygous 6 bp deletion was found in one cystinuric mixed breed dog. According to the genetic breed determination, this dog consisted of 1/4 each from Miniature Poodle, Chihuahua, and Shih-tzu with several other breeds consisting of the last quarter, but with no evidence of Australian Cattle dog. All Australian Cattle dogs, males and females, homozygous or heterozygous for this mutation were cystinuric. Homozygous males showed clinical signs (urethral obstruction) earlier in life than heterozygous males. Thus, cystinuria in Australian Cattle dogs is inherited as an autosomal dominant trait [26]. 

In Miniature Pinschers, a single base change (missense mutation) in exon 9 of the SLC7A9 gene was detected. It caused the substitution of a large positively charged, hydrophilic arginine for the very small, hydrophobic glycine residue. All cystinuric Miniature Pinschers assessed in this study were found to be heterozygous for this mutation. The exonic sequence of the SLC3A1 gene did not indicate any mutation. These results document the heterogeneity in canine cystinuria.

In dogs, cystinuria had been historically divided into two types: type I referring to a mutation in a SLC3A1 gene with autosomal recessive inheritance and non-type I cystinuria which is associated with milder degree of cystinuria and which is observed in mature intact males of various breeds [27]. The new classification system describes type I cystinuria with autosomal recessive inheritance, type II with autosomal dominant inheritance, and type III for sex-limited inheritance. Involvement of the SLC3A1 gene is indicated by adding A, and similarly B indicates mutations in SLC7A9 [26] (Table 1).

Type III cystinuria (androgen-dependent) occurs in intact males only, occurs later in the life when compared to Newfoundlands, is less severe and associated with variable concentrations of cystine in the urine. This type of cystinuria is reported in Mastiffs, French and English Bulldogs, Basset Hounds, and Irish Terriers. An SLC3A1 nonsense mutation appears to be associated with cystinuria in Mastiffs and related breeds, but not in Irish Terriers or Scottish Deerhounds [27]. The underlying cause of this proposed androgen dependency is currently unknown.

Nevertheless, in most affected breeds, the mutation causing cystinuria is not determined. In several breeds, sequencing of the exons of both SLC3A1 and SLC7A9 did not identify any putative underlying mutation [28]. Thus, further studies of various affected breeds are needed to detect the mutation and to determine the mode of inheritance. 

## 3. Cystine Urolithiasis

### 3.1. Solubility of Cystine in Urine

Cystinuria by itself does not result in urolithiasis and many cystinuric dogs do not form uroliths [29]. Major factors involved in urolith formation include supersaturation of urine with calculogenic minerals resulting in crystal formation, effects of urinary inhibitors and promoters of crystallization, crystal aggregation, and growth [30,31]. Cystine uroliths formation is affected mainly by urine pH. Cystine is relatively insoluble at physiological pH levels of 5–7, with a pKa level of 8.3 [32]. Up to pH 7, the solubility of cystine is approximately 250 mg/L, whereas at a urine pH level of 7.5, this will double to 500 mg/L urine and triple at pH 8 or higher [33] (Figure 1).

### 3.2. Prevalence of Cystine Urolithiasis

The prevalence of cystine urolithiasis in dogs varies with geographic location and time (details are specified in Table 2). 

According to the most available reports, the prevalence in North America (United States and Canada) is approximately 1–3% [39,41,44]. The prevalence in European countries is much higher, ranging from 3% up to 26% in older reports or up to 14% in the most recent studies [57,68,69]. Considering the results of recent studies from North America and Europe, the calculated prevalence of cystine urolithiasis is 0.8% in North America and 8.5% in European countries [41,44,51,56,58,60,62,63,65,66,67,68,69]. When evaluating the trends in prevalence of cystine urolithiasis, a gradual increase can be observed in the last decade, both in North America and Europe [41,44,56,68,69]. The latest published report from Minnesota Urolith Center mentioned the cystine urolith prevalence of 7% (from total 61,160 submissions) [72]. Nevertheless, this high number is at least partly affected by submissions from Europe, where the highest portion of cystine uroliths can be seen when compared to other areas.

In all reports on canine cystinuria, males are affected significantly more often than females (98.8% and 1.2%, respectively; these numbers were calculated by taking together available data from scientific reports) [36,37,40,45,47,48,49,50,51,55,56,57,58,59,61,63,65,66,67,68,69,71]. Androgen dependency in type III cystinuria may explain the epidemiological observation that cystine urolithiasis has historically been more common in dogs from European countries than from the USA, where neutering of dogs is more common. In the United States, most (68%) young adult dogs (1–4 years) is castrated and the percentage of castrated dogs gradually increases with age to 81% in adult dogs (4–10 years) and 86% in dogs older than 10 years [73]. In Germany, 43.1% of canine population older than one year is neutered (39% of males and 48% of females) [74]. Similar numbers are reported from England, where 44.73% of male dogs are neutered [75]. Cystine urolithiasis typically occurs in young adult and middle-aged dogs, with reported means from 4 to 6 years [24,37,48,49,52,54,55,66,71].

Various breeds are associated with cystinuria. The most mentioned are English Bulldog, Newfoundland, Dachshund, Chihuahua, Staffordshire Bull Terrier, Rottweiler, French Bulldog, and Miniature Pinscher. Nevertheless, reported breeds vary with the geographical location and time of the study and these results may be affected by the breed popularity. Lulich and Ulrich report more than 170 canine breeds where cystine urolithiasis was diagnosed (without specification) [6]. The list of breeds particularly mentioned in scientific reports is in Table 3.

Cystine uroliths are rarely (3%) reported from the upper urinary tract. The most common localization of cystine uroliths are urinary bladder and urethra. They may cause urethral obstruction with typical clinical manifestation [37,38,55]. This is consistent with the findings in mice or ferrets [12,79] but in contrast with human medicine, where cystine nephroliths are more common [1]. Occasionally, cystine uroliths may be associated with urinary tract infection. Ling et al. (1986) reported the presence of UTI in almost one third of cases [80].

## 4. Diagnosis

### 4.1. Diagnosis of Cystine Urolithiasis

Imaging methods are the most definitive diagnostic tool for detection of urolithiasis in general. They are used to verify the presence of uroliths and their location, number, size, shape, and density [21]. The radiodensity of cystine stones compared to soft tissue is similar to struvite, less than calcium oxalate and calcium phosphate, and greater than urate. Survey radiographs may be insensitive for detection of small cystine uroliths (less than 1 to 3 mm). Double contrast cystography and/or ultrasonography may be needed. However, ultrasonography involves difficulty in detecting uroliths in the ureters and urethra. Thus, the combination of various methods may be necessary [81].

Canine cystine uroliths are usually round or ovoid shape with smooth surface. The color may vary, e.g., yellowish brown, medium-light tan, and a range from light yellow to reddish brown are reported [76,82,83] (Figure 2). They are commonly multiple [82], e.g., Méric et al. reported seven as a median number of stones [65]. Their size varies from less than a millimeter to several centimeters in diameter [76]. Most canine cystine uroliths are pure and few contain other minerals, especially ammonium urate and calcium oxalate or struvite [76,77,78]. In contrast, Escolar et al. reported the presence of small amounts of calcium apatite in at least 55% of canine cystine uroliths [57]. 

An estimation of the urolith composition may be done on the basis of their macroscopic appearance, but this may be associated with considerable errors. Quantitative methods (optical crystallography and infrared spectroscopy) are currently methods of choice [81]. Qualitative analysis showed less than 50% agreement in the case of cystine calculi [84].

### 4.2. Diagnosis of Cystinuria

#### 4.2.1. Urinalysis

Cystine crystals are colorless hexagonal plates. Their six sides may or may not be equal and the crystals tend to aggregate and appear layered (Figure 3). Their detection in urine provides strong support for cystinuria because these crystals do not occur in healthy animals. It is noteworthy that cystine crystals are not constantly present in cystinuric dogs [76].

#### 4.2.2. Assessment of Aminoaciduria

A coloric cyanide-nitroprusside test may be performed. Sodium cyanide reduces cystine to cysteine and the free sulfhydryl groups subsequently react with nitroprusside to form a characteristic purple color [85]. Nevertheless, the test requires dangerous substances. Thus, it is not suitable as an in-house test despite being easy to perform and only selected laboratories offer this test [86].

Direct measurement of urine cystine concentration is the most precise method allowing quantification. The most used techniques are high-pressure liquid chromatography and automated amino acid analyzers. Not all cystinuric dogs show the same pattern of urinary amino acid loss. Some dogs only lose cystine, whereas others demonstraate increased excretion of cystine, as well as ornithine, lysine, and arginine [87]. The difference (isolated cystinuria vs. urinary excretion of other amino acids) may be caused by the genetic background of the disease, i.e., the specific mutation in particular gene and homozygosity or heterozygosity. Genetic variants may affect the impairment of the transmembrane carrier and thus the extent of aminoaciduria. Because of altered tubular reabsorption of the dibasic amino acids associated with cystinuria, the concentration of ornithine, lysine, and arginine should be evaluated together with cystine. The results of COLA amino acids may also support diagnosis in the case cystinuria, because cystine may precipitate and thus cause lower concentrations than were originally present in the urine. The urine concentration of amino acids is expressed as micromoles per gram of creatinine. Dogs with either cystine levels of >200 μmol/g creatinine or COLA values of >700 μmol/g creatinine are considered cystinuric [27]. In cystinuric dogs, the urinary cystine excretion seems to be affected by age. Older dogs over five years were found to have significantly lower cystine levels than younger dogs (five years or younger) [88].

#### 4.2.3. Genetic Tests

In some breeds, genetic tests for cystinuria are available (http://research.vet.upenn.edu/WSAVA-LabSearch, accessed on 1 May 2021). Such tests offer a method of diagnosing cystinuric animals before they present with clinical signs of cystine urolithiasis and may identify not only clinically affected patients, but also asymptomatic carriers. The results may have an impact on breeding programs.

## 5. Treatment and Prevention

Cystinuria per se, as an inborn error of metabolism, cannot be successfully treated. The management of cystinuria is aimed at urolith removal or dissolution in case of urolithiasis and/or prevention of urolith formation. After surgical removal, cystine uroliths commonly recur within 6–12 months [78,89]. Because of the high recurrence rate, prevention is necessary. Without such a strategy, many owners may resort to euthanasia instead of further surgical interventions.

Different therapeutic approaches have been described over the years, such as dietary modification, reduction of urine cystine concentration by induced diuresis, increase of cystine solubility by urine alkalinization and conversion of cystine to a more soluble compound with D-penicillamine or tiopronine [88]. According to the current recommendations on the treatment and prevention of uroliths, medical dissolution should be considered before removal [90]. In cases when dissolution cannot be achieved (medications or dissolution foods cannot be administered or tolerated or the urolith cannot be adequately bathed in modified urine), minimally invasive techniques for urolith removal should be preferred (reviewed in [91]).

### 5.1. Dietary Treatment

Dietary treatment plays a crucial role in the management of cystine stone formation. The dissolution can be achieved by decreasing the concentration of crystallogenic compounds and by increasing cystine solubility.

Urine dilution is an essential step for the prevention and/or dissolution of uroliths regardless of their mineral type. Increased diuresis decreases the concentration of crystal precursors and stimulates more frequent urination, decreasing the time for crystal aggregation [92]. Increasing dietary moisture significantly reduces urinary specific gravity and it is an effective way to enhance diuresis [93].

#### 5.1.1. Protein

High-protein foods should be avoided in dogs at risk of cystine urolithiasis. Consumption of low protein, moist veterinary food led to a 20–25% reduction in 24-h urine cystine excretion in cystinuric dogs when compared to moist, canine adult maintenance food [76]. Urine cystine excretion can be modulated by dietary protein intake, and more specifically methionine (precursor of cysteine) and cysteine. Feeding a diet containing amounts of these essential amino acids close to their minimum is therefore recommended. Most plant protein sources have smaller amounts of sulfur amino acids than animal proteins [21,76]. Protein levels in foods for cystinuric dogs should be between 10% and 18% dry matter [6]. Because of possible taurine and carnitine deficiency, cystinuric dogs should be monitored or their diets should be supplemented with carnitine and taurine [22].

#### 5.1.2. Sodium

Dietary sodium restriction seems to be an important component of the therapeutic strategy in cystinuric people because dietary sodium may enhance cystinuria [94,95]. Dietary sodium in canine therapeutic diets should be limited to less than 0.3% dry matter [6].

#### 5.1.3. Urinary pH

As mentioned above, the solubility of cystine in urine is pH dependent. Beneficial effect has been reported in feeding alkalinizing food. Thus, the food that produces a urinary pH range of 7.1–7.7 is recommended for dogs with cystine urolithiasis. Urinary pH values higher than 7.7 should be avoided until it is determined whether they provide a significant risk factor for formation of calcium phosphate uroliths [6]. When a therapeutic diet alone is not able to provide alkaline urine, the administration of alkalinizing agents is recommended. Because of the reports that dietary sodium may enhance cystinuria, potassium citrate should be preferred to sodium bicarbonate [6]. Desired goals of dietary management are urine pH values above 7.5 and urine specific gravity below 1.020 [86].

### 5.2. Medical Treatment

#### 5.2.1. D-Penicillamine

D-penicillamine (dimethylcysteine) is a first-generation cysteine chelating drug. It interacts with cystine to form a penicillamine-cysteine mixed disulfide in the urine which is 50 times more soluble than free cystine. Consequent decreased free cystine excretion into urine diminishes the likelihood of urolith formation [78,96]. The recommended dose for the treatment of canine cystinuria is 30 mg/kg/day, divided into two subdoses. After oral administration, this drug is rapidly absorbed from the intestine and excreted via kidneys. According to Bovée 1986 [78], the therapy with D-penicillamine is associated with nausea and vomiting in approximately half of treated dogs. Thus, the effectiveness of the medication is limited. The extent of adverse effects is dose dependent. The drug may be mixed with food to prevent vomiting, however this reduces its absorption in gastrointestinal tract [97]. In people, the administration of D-penicillamine is associated with a variety of adverse effects, including glomerulonephropathy with proteinuria, gastrointestinal signs (abdominal pain, diarrhea, vomiting, oral ulcers), hematological abnormalities (thrombocytopenia, leukopenia, aplastic anemia), cutaneous changes (urticaria, pruritus, erythema, alopecia), and dyspnea [98].

Osborne et al. [99] reported fever and lymphadenopathy in a Dachshund given D-penicillamine at a recommended dose. The signs subsided following withdrawal of the drug. Because of a high risk of adverse effects accompanying the treatment and current availability of safer options, D-penicillamine is now not recommended for the management of canine cystinuria [86].

#### 5.2.2. 2-Merkaptopropionyl-glycine (Tiopronin)

Chemically related to D-penicillamine, 2-merkaptopropionyl-glycine (2-MPG, commonly called tiopronin) is a second-generation cysteine chelating agent that decreases the concentration of cystine by a thiol-disulfide exchange reaction. Tiopronin has higher oxidation-reduction potential than penicillamine and may be even more effective [100]. The drug is eliminated almost exclusively by the kidneys with rapid urinary excretion [101].

Tiopronin have been used in the treatment of canine cystinuria since the 1980s and successful dissolution of urolith have been reported [102]. Oral administration of 2-MPG at a daily dose of approximately 40 mg/kg/day divided in two equal doses was effective in complete urolith dissolution in nine of 17 dogs. The daily dose of 30 mg/kg was used as prophylactic and during this course, urolith did not reform in 14 dogs. In four dogs with urolith reformation during the treatment, the uroliths dissolved when the 2-MPG dose was raised back to 40 mg/kg [89]. According to the results of a study evaluating 14 years of clinical experience with the medical treatment of 88 cystinuric dogs, adverse effects were found in 11 dogs. The most severe were aggressiveness towards members of the families and myopathy (bilateral masseter and quadriceps pain, weakness, difficulty chewing and swallowing). The other adverse effects were proteinuria, thrombocytopenia, anemia, high liver enzymes activities and bile acids, tiredness, small pustules of the skin, dry and crusty nose, and sulfur odor of the urine. These signs were noted between one and 36 months (mean 7.6 months) after the start of treatment. All signs gradually disappeared when tiopronin treatment was stopped [88]. Dissolution required 2–4 months of therapy. The combination of litholytic food and 2-MPG therapy is more effective in promoting dissolution of uroliths than either alone. The mean time required to dissolve the cystine uroliths was 78 days (range 11 to 211 days) [103]. Disadvantage of the tiopronin treatment is its high price, which can be deterrent for many owners and inadequate availability because in many countries, tiopronin is not distributed. Current treatment recommendations discourage the use of D-penicillamine and encourage the use of the less toxic 2-MPG [86].

D-penicillamine is well known for its metal-binding properties. The short-term treatment with D-penicillamine conspicuously increased the renal excretion of calcium, copper, and zinc. In contrast, 2-MPG does not to any appreciable extent increase the urinary excretion of metals. Thus, there is no risk for renal losses of biologically important metals [104].

#### 5.2.3. Captopril

Captopril is a thiol-containing angiotensin-converting enzyme inhibitor that is primarily used as an antihypertensive agent. Captopril-cysteine disulfide is 200 times more soluble than cystine. Results of clinical trials suggest that captopril may be clinically efficacious in at least some people with difficulty controlling cystinuria [95]. Currently, there is no report on the use of captopril in canine cystinuria.

#### 5.2.4. Bucillamine

Bucillamine is a drug developed from tiopronin that may have greater affinity for cysteine. It is used as an antirheumatic agent. The efficacy of bucillamine in human cystinuria is currently investigated [95]. Similar to captopril, there are no reports on use of bucillamine in cystinuric dogs.

### 5.3. Castration

Surgical or medical castration can resolve cystinuria in the subset of male dogs with androgen-dependent cystinuria. Castration appears to lower the urinary cystine and COLA concentrations and to prevent cystine calculi formation. The effect of castration in breeds with type of cystinuria seems to have greater effect in comparison with dietary changes [105]. To determine whether neutering reduces cystinuria, measurement of urine cystine concentration before and three months after castration is recommended. If the urine cystine remains elevated at three months, another evaluation should be performed again at six months. Persistently positive results indicate that the dog has a non-androgen-dependent form of the disease [86]. In dogs with androgen-dependent cystinuria, the question may be raised as to whether neutering alone will result in urolith dissolution [90]. In dogs with other types of cystinuria, castration should be recommended as well to prevent further breeding (wanted or accidental) and thus the passing of this condition on to the future generations. 

### 5.4. Future Therapies

#### 5.4.1. L-Cystine Dimethyl Ester and L-Cystine Methyl Ester (L-CDME and L-CME)

A new alternative approach for the prevention of recurrent urolithiasis is based on crystal growth inhibition. It has been shown that L-CDME and L-CME dramatically reduce the growth velocity of cystine molecules [106]. The efficacy of these molecules was demonstrated in vivo using murine models [107]. Nevertheless, further studies are needed to evaluate the effect and safety of this therapy in people or dogs.

#### 5.4.2. Alpha-Lipoic Acid

In a mouse model of cystinuria, it was reported that the nutritional supplement of alpha-lipoic acid inhibits stone formation by increasing the solubility of urinary cystine [108]. Moreover, in this case, clinical trials must be performed.

#### 5.4.3. Selenium

In a double-blinded clinical trial study conducted on 48 humans with cystinuria, selenium supplementation (200 mg/daily for six weeks) led to a significant reduction in the volume of cystine crystals in urine. Therefore, since reducing cystine crystal volume decreases crystal formation, selenium may be effective to cure patients with cystinuria [109]. No similar study has been performed in dogs.

#### 5.4.4. Tolvaptan

Tolvaptan (vasopressin receptor antagonist) showed efficacy in preventing cystine stone growth in cystinuric mice through increased liquid intake and urine volume [110]. The efficacy, short-term safety, and tolerability of tolvaptan was evaluated in a very recent study. Four young patients were enrolled and increased urinary volumes were observed. No abnormalities in serum electrolytes or liver enzymes were found, and only extreme thirst was reported [111]. It is questionable if this type of therapy is suitable for dogs, for whom the opportunity to urinate depends mainly on the owners and their schedule. 

## 6. Cystinuria in Cats

In cats, cystinuria occurs less commonly than in dogs according to the reports of urolith centers. In the United States, feline cystine calculi represents only 0.1% of all feline uroliths (92 in 94 776) compared with 0.75% of canine uroliths (3402 in 451 891) [39]. Recent European studies where canine and feline uroliths were evaluated showed a similar prevalence of feline cystine urolithiasis (0.11% in cats vs. 13.6% in dogs) [57,58,60,69]. Further details are presented in Table 4.

Feline cystinuria was first documented in 1991 in a single case report. A 10-month-old male Siamese cat was referred for cystine crystalluria. In this cat, the fractional reabsorption of cystine, ornithine, lysine and arginine was markedly lower when compared to clinically normal cat [119]. Subsequently, clinical features in 18 cystinuric non-purebred domestic short-haired and purebred cats were summarized. There were eight males (all castrated) and nine females (seven of them spayed); in one cat, the gender was unknown. The mean age of affected cats was 3.6 years with the range from four months to 12.2 years. The cats were presented for signs of lower urinary tract disease. Cystine crystalluria was a characteristic finding. Urine amino acid profiles of four affected cats also revealed increased levels of ornithine, lysine, and arginine. The mode of inheritance was not determined. All uroliths were obtained from the bladder and urethra and were radiodense [120]. The later studies confirmed, that both males and females are affected almost equally, and this is independent of neuter status and thus androgen-dependent type of cystinuria seems to be unlikely in cats [114,118,121].

The first mutation detected in association with cystinuria in cats was the missense mutation in SLC3A1. The affected cat was intact male and was homozygous for this mutation. The cat was presented for signs of lower urinary tract disease with finding of cystine crystals at the age of two months [122]. In a group of seven cystinuric cats, unique SLC7A9 variants were found. All these cats were juvenile to middle-aged when clinical signs first appeared. All cats were either prepubertal or neutered before cystine crystals occurred [121]. In Germany, the missense mutation in SLC7A9 in Siamese-crossbreed littermates was reported [123]. These results show a heterogeneity in cats as reported in dogs or humans. Further studies are needed in this field to obtain more information on genetic background and mode of inheritance. In cats, cystinuria was reported in Domestic Short-haired cat, Domestic medium-haired cat, Domestic long-haired cat, Maine Coon cat, Sphynx, Siamese cat, and Korat [119,120,121,123]. 

The treatment of feline cystinuria is based on the same strategies as the treatment of canine cystinuria. Nevertheless, because of infrequency of this condition, clinical trials are missing. To minimize the recurrence of cystinuria, diet lower in protein and sodium content which produces neutral to alkaline urine should be fed. The urine pH should be above 6.5; if not, potassium citrate can be added. The water intake should be increased by feeding of canned therapeutic food or adding water to food to lower urine specific gravity below 1.030. Because of the lack of clinical trials, the medical treatment of feline cystinuria with 2-MPG should by cautiously considered in recurrent cases [124]. According to the available reports, feline cystinuria seems to occur earlier in the life when compared to dogs [120,122]. Thus, affected cats may be diagnosed earlier and excluded from breeding before they have descendants. This can explain the lower and stable prevalence of feline cystinuria. 

## 7. Conclusions

Despite the fact that cystinuria was the first described inborn error of metabolism in dogs, many questions concerning genetic aspects and mode of inheritance remain. The answers to these questions may help to control cystinuria and decrease its prevalence in the canine population.

## Figures and Tables

**Figure 1 animals-11-02437-f001:**
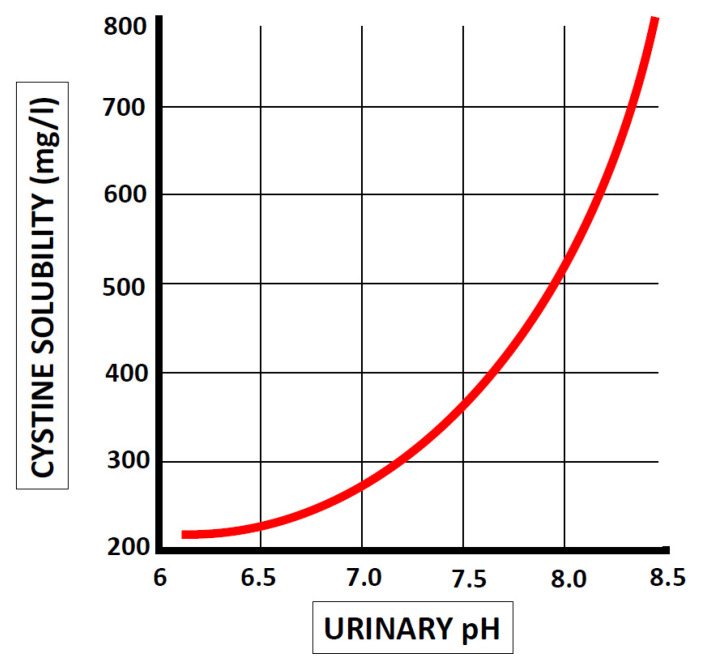
Solubility of cystine at different urinary pH values [34,35].

**Figure 2 animals-11-02437-f002:**
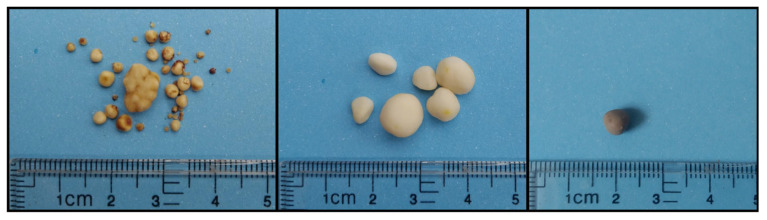
Various canine cystine uroliths obtained via cystotomy. They are usually small, round or ovoid, and light yellow to yellowish brown, tan or reddish brown color.

**Figure 3 animals-11-02437-f003:**
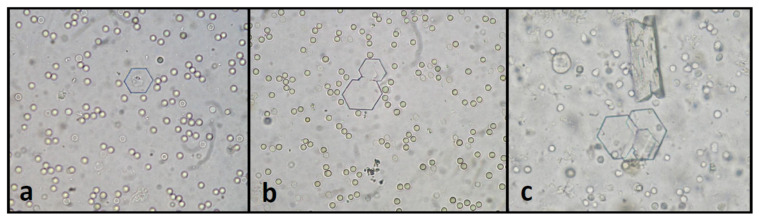
Canine cystine crystals as colorless hexagonal plates in unstained urine sediment ((**a**,**b**)–surrounded by erythrocytes, (**c**)–surrounded by leukocytes and struvite crystal).

**Table 1 animals-11-02437-t001:** Classification system for cystinuria according to Brons et al. 2013 [26].

Phenotype		Type I-A	Type II-A	Type II-B	Type III
**Mode of inheritance**		Autosomal recessive	Autosomal dominant	Autosomal dominant	Sex limited
**Gene**		SLC3A1	SLC3A1	SLC7A9	Undetermined
**Sex**		Males and females	Males and females	Males and females	Intact adult males
**Androgen dependence**		No	No	No	Yes
**COLA** **μmol/g creatinine ***	**homozygous**	≥8000	≥8000		≤4000
	**heterozygous**	≤500	≥3000	≥700	
**Breeds**		Newfoundland Landseer Labrador	Australian Cattle Dog	Miniature Pincher	Mastiff and related breeds Scottish Deerhound Irish Terrier

* normal ≤ 500 μmol/g creatinine.

**Table 2 animals-11-02437-t002:** Prevalence of cystine urolithiasis in dogs by region and characteristics of affected dogs in various studies.

Location	Author	Method of Analysis	Years	Total Number	Cystine Uroliths	Sex	Age	Breeds
**America**								
United States	Ling and Ruby (1986) [36]	quantitative	1981–1984	813	21 (2.6%)	20 males 1 female		
United States	Case et al. (1992) [37]	crystallography	1981–1989	5375	107 (2.0%)	106 males 1 female	mean 4.5 years	Australian Cattle Dog, Mastiff, English Bulldog
United States	Osborne et al. (1999) [38]	quantitative including infrared spectroscopy	1981–1997	77,191	760 (1%) ^a^			
United States	Osborne et al. (2009) [39]	quantitative including infrared spectroscopy	1981–2007	451,891	3 402 (0.8%)			
United States	Low et al. (2010) [40]	crystallography	1985–2006	25,499	320 (1.3%)	313 males 7 females		English Bulldog (OR 44.2), Newfoundland (OR 12.6), Dachshund (OR 7.6), Chihuahua (OR 5.6), Miniature Pinscher (OR 9.3), Welsh Corgi (OR 5.0) ^b^
United States	Kopecny et al. (2021) [41]	quantitative	2006–2018	10,444	279 (2.7%)	273 males (192 intact males) 5 females		Mastiff (OR 52.7), Australian Cattle Dog (OR 30.8), Pitbull Terrier (OR 12.9), Rottweiler (OR 11.9), English Bulldog (OR 10.1), Bulldog (OR 9.1) ^c^ Females: Pitbull Terrier, crossbreed, Newfoundland
Canada	Houston et al. (2004) [42]	crystallography (+another quantitative methods) ^d^	1998–2003	16,647	59 (0.4%)	58 males 1 female	mean in males 4.3 years Female–4 years old	English Bulldog, Newfoundland, Chihuahua, Rottweiler, Scottish Deerhound
Canada	Houston and Moore (2009) [43]	crystallography (+ another quantitative methods) ^d^	1998–2008	40,637	115 (0.3%) ^e^			
Canada	Houston et al. (2017) [44]	crystallography (+ another quantitative methods) ^d^	1998–2014	79,965	480 (0.6%) ^f^	significantly more frequent in males		Scottish Deerhound, Whippet, Newfoundland
Mexico	Del Angel-Caraza et al. (2010) [45]	quantitative		105	1 (1%)	male	4–6 years	
Mexico	Mendoza-Lopez et al. (2019) [46]	quantitative		195	0			
**Europe**								
UK	White (1966) [47]	chemical methods	1st series 1944	103	18 (18%)	males		Corgi, Dachshund
			2nd series 1961–1966	737	114 (15.5%)	males		
UK (Scotland)	Weaver (1970) [48]	chemical methods	1961–1968	100	20 (20%)	males	mean 5.3 years	Basset Hound, Irish Terrier
UK	Clark (1974) [49]	X-ray diffraction		110	24 (22%)	males	4.9 ± 2.03 years	
UK	Allen et al. (2008) [50]	quantitative ^g^	2002–2006	11,027	348 (3.2%)	347 males 1 female	mean 73 months	Staffordshire Bull Terrier
UK	Rogers et al. (2011) [51]		2002–2010	5591	180 (3.2%)	males		
UK	Roe et al. (2012) [52]	quantitative ^g^	1997–2006	14,008	424 (3%)	more common in males	majority at the age 36–72 months	English bulldog (OR 60.88), Staffordshire Bull Terrier (OR 8.71), Rottweiler (OR 6.99), Jack Russel Terrier (OR 2.32) ^h^
Germany	Hesse (1990) [53]			1731	387 (22.4%)			Dachshund, Munsterlander, Irish Terrier
Germany	Hesse et al. (2012) [54]		1979–2007	15,494	1491 (9.9%) ^i^	1476 males 15 females	6.0 ± 2.5 years	Dachshund, Dobermann Pinscher, Poodle, Cocker spaniel, Schnauzer, Yorkshire Terrier, Pekingese, Shih-tzu, Dalmatian
Germany	Hesse et al. (2016) [55]		1979–2013	20,316	1760 (8.7%)	1741 males 19 females	5.9 ± 2.5	
Germany	Breu et al. (2021) [56]		2017–2019	2772	421 (15.2%)	Males: 324 intact, 61 castrated Females: 6 intact 4 castrated ^j^	median 5 years	French Bulldogs, Bulldogs, Chihuahua, Dachshund
Spain	Escolar et al. (1990) [57]			171	44 (26%)	males		
Spain	Riesgo et al. (2018) [58]	quantitative ^g^	2004–2017	116	9 (7.8%)	males	2–12 years	Basset Hound
Spain and Portugal	Vrabelova et al. (2011) [59]	quantitative ^g^	2004–2006	2765	87 (3%)	86 males 1 female		Bulldogs
Portugal	Tomé et al. (2007) [60]	quantitative ^g^	2004–2006	299	20 (6.7%)			
Czech Republic	Sosnar et al. (2005) [61]	infrared spectroscopy	1997–2002	1366	77 (5.6%)	45 males ^k^		Dachshund Basset Hound
Czech Republic	Kučera and Kořistková (2017) [62]	infrared spectroscopy	2003–2016	803	41 (5.1%)			
Romania	Mircean et al. (2006) [63]	infrared spectroscopy	2005–2006	20	2 (10%)	males		
France	Blavier et al. (2012) [64]	infrared spectroscopy	2007–2010	1131	42 (3.7%)			
France	Méric et al. (2020) [65]			2054	183 (8.9%)	182 males1 female		English Bulldog, American Staffordshire Terrier, French Bulldog, Staffordshire Bull Terriers, Dachshunds
Hungary	Bende et al. (2015) [66]	infrared spectroscopy	2001–2012	2543	108 (4.2%)	96 males ^l^	58 ± 31.3 months	Basset Hound (OR 40.2), Bulldog (OR 18.6), Rottweiler (OR 13.9), Min. Pinscher (OR 12.7), Wirehaired Dachshund (OR 7.6), Dachshund (OR 6.5), Chihuahua (OR 4.8) ^m^
Switzerland	Brandenberger-Schenk et al. (2015) [67]	quantitative ^g^	2003–2009	490	17 (3%)	males	median 3.9 years (range 0.6–10.1)	English Bulldog
Norway	Lund and Thoresen (2020) [68]		2010–2019	684	97 (14.2%) ^n^	Males: 91 intact, 2 castrated Females: 3 intact 1 castrated		
The Netherlands	Burggraaf et al. (2021) [69]	quantitative	2014–2020	4369	601 (13.8%)	593 males (455 intact, 138 neutered) 8 females (2 intact, 6 neutered)		American Staffordshire Terrier, Basset Hound, Chihuahua, English Bulldog, French Bulldog, Miniature Pinscher, Rottweiler, Dachshund, Yorkshire Terrier
**Asia and Oceania**								
New Zealand	Jones et al. (1998) [70]	X-ray diffraction	1993–1996	316	24 (7.6%)			
Thailand	Hunprasit et al. (2017) [71]	quantitative ^g^	2009–2015	8560	136 (1.6%)	126 males 2 females ^o^	4.8 ± 2.4	Chihuahua, French Bulldog, Shih-tzu, Miniature Pinscher

^a^ The prevalence of cystine urolithiasis decreased during the period. ^b^ OR = odds ratio. Odds ratio was calculated by logistic regression analysis by comparing breed distributions in dogs with cysteine urolithiasis with breed distributions of 2 groups (dogs with other urolith types and dogs examined at the Veterinary Medical Teaching Hospital at the University of California during the same period as the study). ^c^ OR = odds ratio. Odds ratio was calculated by logistic regression analysis by comparing breed distributions in dogs with individual urolith type to mixed breed dogs with the same mineral type. ^d^ X-ray microanalysis, infrared spectroscopy, scanning electron microscopy. ^e^ Significant decrease of cystine urolith prevalence during the study period. ^f^ Significant increase of cystine urolith prevalence during the study period. ^g^ Uroliths were analyzed in Minnesota Urolith Centre. ^h^ OR = odds ratio. Chi-squared tests were performed to assess whether particular breeds were over-represented among the dogs forming cystine uroliths compared with the national insurance company database. ^i^ The prevalence of cystine gradually decreased from 27% between the years 1979–1985 to 5.5% in period from 2000 to 2007. ^j^ In 26 cases, the sex was not reported. ^k^ Sex was reported in 45 cases only. ^l^ In 12 cases, the sex was not provided. ^m^ OR = odds ratio. Odds ratio was calculated by logistic regression analysis by comparing of the dogs with cystine uroliths to general population of dogs in Hungary according to the Hungarian Microchip Register. ^n^ A gradual increase in cystine uroliths was noted (from 12% in 2010 to 30% in 2018). ^o^ In 6 cases, the sex was not reported.

**Table 3 animals-11-02437-t003:** The list of canine breeds where cystinuria was reported [20,36,37,44,47,48,55,56,57,65,70,76,77,78].

Canine Breeds		
Afghan	French Bulldog	Pug
Akita Inu	German Braque	Puli
Alaskan Malamute	Golden Retriever	Rat Terrier
American Staffordshire Terrier	Great Dane	Rottweiler
Australian Cattle Dog	Husky	Rough Collie
Australian Shepherd Dog	Chihuahua	Saluki
Australian Terrier	Irish Terrier	Samoyed
Basenji	Jack Russel Terrier	Scottish Deerhound
Basset Hound	Kromfohrländer	Scottish Terrier
Bichon Frise	Labrador Retriever	Setter
Border Collie	Landseer	Shetland Collie
Borzoi	Lhasa Apso	Shetland Sheepdog
Boxer	Maltese	Shih Tzu
Brussels Griffon	Mastiff	Schnauzer
Bull Mastiff	Miniature Pinscher	Silky Terrier
Cairn Terrier	Miniature Poodle	Staffordshire Bull Terrier
Cavalier King Charles Spaniel	Miniature Schnauzer	Staffordshire Terrier
Cocker Spaniel	Munsterlander	Swedish Lapphund
Dachshund	Newfoundland	Tibetian Spaniel
Dalmatian	Old English Sheepdog	Welsh Corgi
Dobermann	Pekingese	West Highland White Terrier
Drever	Pitbull Terrier	Whippet
English Bulldog	Pointer	Yorkshire Terrier
Fox Terrier	Poodle	

**Table 4 animals-11-02437-t004:** Prevalence of cystine urolithiasis in cats by region and characteristics of affected cats in various studies.

Location	Author	Years	Total Number	Cystine Uroliths	Sex	Age	Breeds
**America**							
United States	Osborne et al. (1984) [112]		328	0			
United States	Osborne et al. (1996) [113]		9481	26 (0.3%)	17 males 6 females	3.2 years (range 4 months–11 years)	DSH, DLH, Siamese, Korat
United States	Cannon et al. (2007) [114]	1985–2004	5230	7 (0.1%)	3 males 4 females		4× DSH
United States	Osborne et al. (2009) [39]	1981–2007	94,776	92 (0.1%)			
United States	Kopecny et al. (2021) [41]	2005–2018	3940	2 (0.05%)			
Canada	Houston et al. (2003) [115]	1998–2003	4866 uroliths 618 urethral plugs	5 uroliths (0.1%) 1 plug (0.2%)			
Canada	Houston et al. (2009) [43]	1998–2008	11,353	11 (0.1%)			
Canada	Houston et al. (2016) [116]	1998–2014	21,426	20 (0.1%)			
**Europe**							
Spain	Escolar et al. (2003) [57]		34	0			
Portugal	Tomé et al. (2007) [60]	2004–2006	65	0			
Switzerland	Schenk et al. (2010) [117]	2002–2009	855	2 (0.2%)			
Spain	Riesgo et al. (2018) [58]	2004–2017	21	0			
The Netherlands	Burggraaf et al. (2021) [69]	2014–2020	3497	4 (0.1%)	3 males 1 female		
**Asia and Oceania**							
New Zealand	Jones et al. (1998) [70]	1993–1996	53	0			
Thailand	Hunprasit et al. (2019) [118]	2010–2017	923	7 (0.8%)	4 males 3 females		6× DSH 1× Persian

DSH–Domestic Short-haired cat, DLH–Domestic Long-haired cat.

## Data Availability

Not applicable to the present work.

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
