# Peer review of "Cystinuria in Dogs and Cats: What Do We Know after Almost 200 Years?"

_animals, 2021, doi:10.3390/ani11082437_

Round 1

Reviewer 1 Report

This is an excellent and comprehensive review of cystinuria and cystine stone formation in dogs and cats.

Minor comments:

  1. It is necessary to explain COLA in the Abstract.
  2. Page 1, line 34: Replace “…these dibasic...” with “...the other dibasic...”
  3. I am personally not particularly happy with the unit µmol/g creatinine, it had been much better to also express creatinine in SI-units.
  4. It is not until late in the manuscript that the reader is informed that the most common location for cystine stones is in the lower urinary tract, bladder and urethra. It had been a great advantage if the references given in Table 2 also included some information on where the stones were located.
  5. Is CT not an appropriate diagnostic tool in dogs and cats?
  6. In the second paragraph on page 10 I would like to see SI-units as well and if possible, also the concentrations of cystine expressed as mmol/L or µmol/L.

Author Response

Dear reviewer,

thank you very much for your comments which improve our work.

Point 1. It is necessary to explain COLA in the Abstract.

Response 1: Explained.

Point 2. Page 1, line 34: Replace “…these dibasic...” with “...the other dibasic...”

Response 2: Replaced.

Point 3: I am personally not particularly happy with the unit µmol/g creatinine, it had been much better to also express creatinine in SI-units.

Response 3: The units used in this manuscript follows the latest scientific works, where the urine cystine concentration is reported (Giger et al. 2011, Brons et al. 2014, Mizukami et al. 2016 and others). In our opinion, the use of other units might be confusing for the reader.

Point 4: It is not until late in the manuscript that the reader is informed that the most common location for cystine stones is in the lower urinary tract, bladder and urethra. It had been a great advantage if the references given in Table 2 also included some information on where the stones were located.

Response 4: Most of reports cited in Table 2 deals with total numbers of all mineral types of uroliths country and period. Only minority (4/36) of them mentioned the location of cystine uroliths. The table 2 is quite big now and adding a new column may decrease the comprehensibility. Thus, this information is in text.

Point 5: Is CT not an appropriate diagnostic tool in dogs and cats?

Response 5: CT as advanced imaging technique may be useful in diagnosis of canine and feline cystinuria, nevertheless, the diagnosis is usually obtained using ultrasound or X-ray which do not need anesthesia and are cheaper.

Point 6: In the second paragraph on page 10 I would like to see SI-units as well and if possible, also the concentrations of cystine expressed as mmol/L or µmol/L.

Response 6: As above.

Sincerely,

Authors

Reviewer 2 Report

This review manuscript by Kovarikova et al. “Cystinuria in dogs and cats: What do we know after almost 200 years?” is a comprehensive study of the genetic disease cystinuria in dogs and cats. The review does a fantastic job of summarizing current knowledge. This review will be of use to veterinarians who may encounter cystinuria in pets as well as those scientists developing treatment strategies for cystinuria or performing research on cystinuria. The manuscript is well organized and is a scholarly resource that will be valuable to others, but there are a few English mistakes so it would benefit from another pass of careful editing, perhaps by an editing service. Suggestions that may improve the manuscript are below.

Major suggestions:

  1. In your discussion on page 8 line 181, compare to stone location for other species. For instance mice, like dogs and cats, develop cystine uroliths in bladder whereas humans develop kidney stones.
  2. Figure legends are far too short and should be more descriptive.
  3. Page 10 Line 230, expand on why do some dogs only lose cystine whereas others lose the COLA AAs?
  4. Page 13 Line 372 – Should not any dog with cystinuria be castrated if for no other reason than to prevent the genetic disease from being passed on, in addition to the possible therapeutic benefits? Would there be any downside to castration? This is a short section that should be expanded.
  5. Page 13 Line 373 – 383. Future Therapies section could be expanded by covering more recent studies in animal models.
  6. The section on cats is quite short compared to dogs, so if there are some means to expand it would help to make the discussion more even. There are many fewer studies but seems that cystinuria prevalence is more equal between the US and Europe for cats. Is the rate of neutering for cats more similar between the continents than it is for dogs? Is the lower rate of cystinuria due to the lower population of purebred cats?

Minor suggestions:

  1. Please reply in your response to reviewers with the contributions of each author to the study and how author order was decided. Why is the least established author listed last?
  2. Add some quantitative #s to the abstract to back up the statements and add the differences between U.S. and Europe (expand on castration).
  3. The definition of COLA on page 1, line 33 is too confusing for the uninitiated since the C is implied from the previous sentence.
  4. Page 2 Line 66 makes it sound like 12 transmembrane domains are typical rather than that the transmembrane domains itself are typical.
  5. Page 2 Line 70, “nonsense” not “non-sense”
  6. Page 2 Line 75, “having” not “heaving”
  7. Page 2 Line 85, remove comma
  8. Page 2 Line 89, codon GGC is not specific enough, state the residue #
  9. Page 3 Line 107, “in” should be “as”
  10. Page 3 Table 1, the formatting could be improved, especially the part for “COLA, homozy-gous and heterozy-gous”
  11. Page 4 Line 133, the “way” of inheritance, consider “method” or “type”
  12. Figure 1 is very large and the curve is pixelated and lacking in data points, can it be improved?
  13. Table 2, the formatting could be improved because it is hard to see where the divisions are by continent and the header region is in a strange place on pages 6 and 7. Consider adding “Prevalence of cystine urolithiasis” ‘by country’ or ‘by region’ or similar since the table is organized this way.
  14. Page 8 Line 168, consider “most young adult dogs (1-4 years) are castrated and the percentage of castrated dogs” or similar change
  15. Page 10 Line 248, consider rewording this sentence.
  16. Page 13 line 378, italicize “in vivo”

This is a great review, thank you for writing it.

Author Response

Dear reviewer,

thank you very much for your comments which improve our work. We were trying to cope with all of them.

Point 1. In your discussion on page 8 line 181, compare to stone location for other species. For instance mice, like dogs and cats, develop cystine uroliths in bladder whereas humans develop kidney stones.

Response 1: This information was added.

Point 2: Figure legends are far too short and should be more descriptive.

Response 2: The legends are more precise now.

Point 3: Page 10 Line 230, expand on why do some dogs only lose cystine whereas others lose the COLA AAs?

Response 3: We added possible explanation into the manuscript  - genetic background, specific mutation in particular genes, heterozygosity/homozygosity are mentioned.

Point 4: Page 13 Line 372 – Should not any dog with cystinuria be castrated if for no other reason than to prevent the genetic disease from being passed on, in addition to the possible therapeutic benefits? Would there be any downside to castration? This is a short section that should be expanded.

Response 4: Thank you very much for this question. You are right and this section was expanded.

Point 5: Page 13 Line 373 – 383. Future Therapies section could be expanded by covering more recent studies in animal models.

Response 5: The treatment with selenium and tolvaptan was added into this section.

Point 6: The section on cats is quite short compared to dogs, so if there are some means to expand it would help to make the discussion more even. There are many fewer studies but seems that cystinuria prevalence is more equal between the US and Europe for cats. Is the rate of neutering for cats more similar between the continents than it is for dogs? Is the lower rate of cystinuria due to the lower population of purebred cats?

Response 6: We tried to expand this part of the manuscript. We think, that the prevalence of feline cystinuria is lowr whned compared to dogs because of earlier onset of clinical symptoms (even in juvenile age). It allows earlier diagnosis and exclusion of affected cats from breeding before they have any descendants. Androgen-dependent cystinuria is less probable in cats (males and females are affected almost equally).

Minor:

Point 1: Please reply in your response to reviewers with the contributions of each author to the study and how author order was decided. Why is the least established author listed last?

Response 1: The order of authors of this manuscript is a result of collective consensus. The contribution of Ms. Vrbová was very valuable.

Point 2-11, 14-16: Corrected

Point 12: Figure 1 is very large and the curve is pixelated and lacking in data points, can it be improved?

Response 12: In case of the Figure 1, we see standard size with no abnormalities.

Point 13: Table 2, the formatting could be improved because it is hard to see where the divisions are by continent and the header region is in a strange place on pages 6 and 7. Consider adding “Prevalence of cystine urolithiasis” ‘by country’ or ‘by region’ or similar since the table is organized this way.

Response 13: We tried to improve the formatting of all tables.

Sincerely,

Authors